# Diversity and Co-Occurrence Pattern Analysis of Cecal and Jejunal Microbiota in Two Rabbit Breeds

**DOI:** 10.3390/ani13142294

**Published:** 2023-07-13

**Authors:** El-Sayed M. Abdel-Kafy, Kamel I. Kamel, Marco Severgnini, Shama H. A. Morsy, Paola Cremonesi, Shereen S. Ghoneim, Gabriele Brecchia, Neama I. Ali, Yasmein Z. Abdel-Ghafar, Wael A. H. Ali, Hoda M. A. Shabaan

**Affiliations:** 1Animal Production Research Institute (APRI), Agricultural Research Center (ARC), Dokki, Giza 12651, Egypt; ki_kamel@yahoo.com (K.I.K.); shwshwhsny909@gmail.com (S.H.A.M.); shereenghoneim25@yahoo.com (S.S.G.); jasmein2004@hotmail.com (Y.Z.A.-G.); waha200@yahoo.com (W.A.H.A.); hodamshabaan@yahoo.com (H.M.A.S.); 2Institute of Biomedical Technologies, National Research Council (ITB-CNR), Via f.lli Cervi, 93, 20054 Segrate, Italy; marco.severgnini@itb.cnr.it; 3Institute of Agricultural Biology and Biotechnology, National Research Council (IBBA-CNR), Via Einstein s/n, 26900 Lodi, Italy; paola.cremonesi@ibba.cnr.it; 4Department of Veterinary Medicine, University of Milano, Via dell’Università 6, 26900 Lodi, Italy; gabriele.brecchia@unimi.it; 5Department of Cell Biology, Biotechnology Research Institute, National Research Centre, Dokki, Giza 12622, Egypt; nehma_6@yahoo.com

**Keywords:** rabbits, microbiota, 16S rRNA gene sequencing, jejunum, cecum, breeds, growth, NMER breed, Giant Flander breed

## Abstract

**Simple Summary:**

A large population of bacteria, protozoa, fungi, and algae colonizes an animal’s body surface. The complex community of microbes that occupies a specific area of the organism and develops symbiotic relationships with the host is referred to as the microbiota. In this context, the gut microbiota plays an important physiological role as it influences the digestion and absorption of nutrients, the development and maturation of the immune system, and thus the growth, resistance to disease, and welfare of the animal. For these reasons, it is important to know the role of the microbiota in these specific functions, as well as which factors can impact the composition of bacterial populations, because changes in the microbiota can result in both beneficial and detrimental effects (dysbiosis) on the host. This study, in addition to providing information on the composition of the microbiota in the jejunum and cecum of rabbits, also evaluates the effect of breed and different growth rates using a modern methodological approach. Since the effect of these factors on the modulation of the gut microbiota has been little studied, this research could be the starting point for new experimental investigations aimed at enhancing rabbit productivity and welfare.

**Abstract:**

This study aimed to evaluate the productive performance and microbiota variation in the jejunum and cecum of two rabbit breeds with different growth rates. This study was carried out on Native Middle-Egypt Breed (NMER) and Giant Flanders (GF) rabbits from 5 weeks to 12 weeks of age. Twenty NMER (NM) and GF male rabbits were slaughtered, and the jejunum and cecum tracts were collected to assay gut microbiota composition via 16S ribosomal RNA (rRNA) gene sequencing and histology examination. At 12 weeks of age, daily weight gain, villus height in the jejunum, total protein, and albumin were higher in GF rabbits than in NMER rabbits. Also, the jejunal villi of GF were well arranged in their dense borders. The microbiota between the jejunum and cecum was significantly different in terms of Beta-diversity. A significant correlation between *Enterococcus* (jejunum NM samples) and *Lactobacillus* (cecum GF samples) with body weight and weight gain was found (*p* < 0.05). Moreover, Escherichia-Shigella in the cecum of NM was significantly correlated with weight gain (*p* < 0.05). The most abundant genera identified in the jejunal and cecal contents of GF were generally beneficial microbiota. They may also play a role in reducing the pathogenic effects of *Escherichia coli* in these rabbits.

## 1. Introduction

The microbial population that inhabits specific niches in the body, developing symbiotic relationships with the host, is called the microbiota. This complex ecosystem of microbial communities is involved in important physiological processes in the host, such as nutrient digestion and absorption [1], immune system development [2], intestinal permeability [3], and both direct and indirect protection against the development of dangerous or pathogenic bacteria [4]. The gut microbiota plays an important role in healthy animals but can also be a cause of pathology [5]. In recent decades, the development of high-throughput sequencing technologies, such as 16S ribosomal RNA (rRNA) amplicon sequencing, has reduced execution costs and facilitated knowledge of the microbiota composition of the different organs of the gastrointestinal tract (stomach, small, and large intestine) in various animal species such as pigs [6], poultry [7], and horses [8]. Also, 16S rRNA sequencing can be used to study the changes induced by managerial, environmental, and individual factors in microbial populations [9,10,11]. Recently, this technology has also been used in domestic rabbits (*Oryctolagus cuniculus*) that are monogastric, hindgut-fermenting herbivores and rely on cecotrophy. The rabbit is a very interesting species because it is considered, at the same time, a farm animal, a pet, and also a laboratory animal [12]. The composition of the microbiota of different intestinal tracts has been evaluated in livestock [13], laboratory [14], pet, and shelter rabbits [15], finding a different microbiota composition in the diverse gastrointestinal tracts. In particular, the large intestine shows the highest richness and diversity in bacterial species, while the small gut has the highest variability in the gastrointestinal tract of rabbits [13]. Several studies have been performed to evaluate the effects of the quality and levels of the diet [16,17], dietary supplementation with nutraceuticals [18,19,20], the temperature of the drinking water [21], age [22], and weaning period [23]. The fecal microbiota and its functional capacity associated with weaning weight in meat rabbits [24], hygiene of the environment [25], season [15], and drug treatments [26,27] have been investigated. It was established that these factors influencing the microbiota composition can also affect the quality of the carcass and meat of the rabbit [28,29,30]. Actually, only a few studies evaluated the effect of the breeds and the growth rate on the microbiota composition [31]. Significant differences in the gut microbiota observed in two rabbit breeds and some families, Ruminococcus and Lachnospiraceae, could be considered biomarkers for improving the health and production performance of meat rabbits [31]. The present study aimed at comparing, using a 16S rRNA-based analysis, the microbial populations in the jejunum and cecum of two breeds of rabbits: the Native Middle Egypt Rabbit—NMER (NM) as a local breed with a light body weight and the Giant Flander (GF) as an exotic breed with a heavy body weight. In addition, histology, scanning electron microscope (SEM) examinations, and biochemical parameters were used to compare NM with GF rabbits.

## 2. Materials and Methods

### 2.1. Experimental Design and Sampling

The experimental protocols were approved by the Animal Production Research Institute (APRI)’s animal care and use committee (ethical approval number: 2021920153429). The current research was conducted by APRI, the Agricultural Research Center (ARC), at the Rabbitry Research Farm in Sakha, Kafr el-Sheikh Governorate, which is located in the north of Egypt, during the period from June to September 2020.

A total of 200 Native Middle-Egypt Rabbit-NMER (NM) and Giant Flanders (GF) breeds, NM (n = 107) and GF (n = 93), rabbits in growing stages (5–12 weeks) were used in the present study. At 5 weeks of age, the young rabbits were weaned and housed in individual cages with 35 × 35 × 25 cm dimensions and equipped with feeding hoppers made of galvanized steel and nipples for automatic drinking. The animals were bred under the same environmental conditions and were watered and fed ad libitum from a commercial pelleted diet without probiotics or antibiotics. The ingredients and chemical composition of the diet used in the experiment are presented in Table 1.

At 12 weeks, 20 male rabbits of the NM and GF breeds with similar weight at weaning (from 490.5 g to 542.5 g, average = 516.5 g) were selected for the study to avoid the effect of sex on studied traits. The rabbits were euthanized at 12 weeks. At euthanasia, the gastrointestinal tracts were removed shortly after death, and the contents of the jejunum and cecum were collected and immediately stored at −80 °C until genomic DNA extraction.

### 2.2. Blood Biochemical Parameters

Before euthanasia, blood samples were collected from all the rabbits (n = 10 for each breed) and centrifuged at 1500× *g* for 20 min; the serum samples were then kept at −80 °C for further biochemical parameter analyses: glucose, total protein, albumin, globulin, triglycerides (TG), and urea were analyzed according to the manufacturing instructions of the Biodiagnostic company kits (Dokki, Giza, Egypt; www.bio-diagnostic.com, accessed on 22 March 2021). The total protein and albumin differences in the collected samples were used to determine the globulin levels. The biochemical parameters were measured using a UV-VIS spectrometer (model T60UV, PG Instruments Limited, Lutterworth, UK).

### 2.3. Histological Characteristics

Immediately after the excision, the samples from the jejunum and cecum (n = 10 for each breed, NM and GF) were preserved in neutral buffered formalin. Then, the tissues were dehydrated in an ascending grade of ethanol and embedded in paraffin wax. Serial sections were cut at 5 μm with a microtome (Galileo SEMI, Diapath, Martinengo, Italy). The cross-sections of both the jejunum and cecum—three sections for each organ per rabbit—were stained with Mayer’s hematoxylin and eosin (H&E). Histological morphometric characteristics in the jejunum and cecum included villus height (VH), villus width (VW), and tunica muscularis (TM), according to [32]. Ten individual villus and ten loci in the tunica muscularis were measured for each rabbit. Histological characteristics were studied using a Leica light microscope and imaging software from Leica Microsystems (Application Suite 3.1.0 software, Leica, Wetzlar, Germany).

### 2.4. Electron Microscopic Examination

The collected jejunum and cecum samples (n = 10 for each breed, NM and GF, and one sample for each organ) were fixed in 3% glutaraldehyde and 0.1 M sodium cacodylate buffer (pH 7.0) for 2 h at room temperature, then rinsed in the same buffer, and finally post-fixed in 1% osmium tetroxide for another 2 h at room temperature. The scanning electron microscope (SEM) samples were dehydrated in an ethanol series ranging from 10% to 90% for 15 min in each alcohol dilution, followed by 30 min in absolute ethanol. The SEM samples were critical-point dried by using liquid carbon dioxide. Specimens were mounted on aluminum stubs with silver paint and coated with gold/palladium in a SPI-Module Sputter Coater device (SPI Supplies, West Chester, PA, USA). The SEM was performed in the electron microscope JEM-2100 (JEOL, Ltd., Tokyo, Japan) at a 20 kV accelerating voltage for studying the villi of the cecum and jejunum.

### 2.5. DNA Extraction, Library Generation, and Sequencing

Total bacterial genomic DNA was isolated from the content of the jejunum and cecum (n = 10 for each breed, NM and GF) by using the Easy Pure Stool genomic kit protocol (TransGen Biotech, Haidian District, Beijing, China) following the manufacturer’s instructions. The 16S ribosomal RNA (rRNA) gene was amplified using primers targeting the V3-V4 hypervariable regions according to the 16S Metagenomic Sequencing Library Preparation (https://support.illumina.com/documents/documentation/chemistrydocumentation/16s/16s-metagenomic-library-prep-guide-15044223-b.pdf, accessed on 3 July 2023, Illumina, San Diego, CA, USA). All of the PCR amplifications were performed in 25 µL. A total of 12.5 µL of KAPA HIFI Master Mix 2× (Kapa Biosystems, Inc., Wilmington, MA, USA) and 0.2 µL of each primer (100 µM) were added to 2 µL of genomic DNA (5 ng/µL). Blank controls (no DNA template added to the reaction) were also performed. The PCR conditions were as follows: 94 °C for 5 min, 98 °C for 30 s, 56 °C for 60 s, and 72 °C for 60 s for a total of 25 cycles, with a final extension step at 72 °C for 10 min.

Amplicons were cleaned with Agencourt AMPure XP (Beckman, Coulter Brea, CA, USA), and libraries were prepared following the 16S Metagenomic Sequencing Library Preparation Protocol (Illumina, San Diego, CA, USA). The libraries obtained were quantified by real-time PCR with KAPA Library Quantification Kits (Kapa Biosystems, Inc., Wilmington, MA, USA), pooled in equimolar proportion, and sequenced in one MiSeq (Illumina) run with 2 × 250-base paired-end reads.

#### 2.5.1. Bioinformatics—Sequence Processing

Raw paired reads from each sample were merged into one single sequence per fragment by PandaSeq [33]; then, low-quality bases (Phred quality score < 3) were trimmed from the 3′-end and fragments having a length < 75% of the initial fragment length were discarded. Filtered reads were clustered into zero-radius operational taxonomic units (zOTUs) by USEARCH (v. 11.0.667, Edgar), retaining only those supported by 5 or more reads. All downstream analyses were performed in the QIIME 1.9.0 suite [34]. Taxonomic assignment of zOTUs was performed by the RDP classifier [35] against the SILVA 132 database [36], using 0.5 as a confidence threshold.

The dataset comprised a total of 40 samples, derived from 2 organs (jejunum and cecum) and 2 rabbit breeds (NM and GF). Each combination had 10 independent replicates. The comparison among the experimental categories comprised three levels of analysis: (a) Breed: comparison between microbiota from GF and NM rabbits (as a whole, considering both organs); (b) Organ: comparison between microbiota from caecum and jejunum (as a whole, considering both breeds); and (c) stratified analyses comparing organs within the same breed and breeds in samples from the same organ.

#### 2.5.2. Statistical Analysis Diversity

In order to have a comparable representation of the bacterial communities, read counts per sample were normalized to the least-sequenced one, at 18,561 reads per sample. The analysis of the sample biodiversity (i.e., alpha-diversity) was based on different metrics (i.e., Shannon’s diversity, chao1 diversity index, observed species, and Faith’s phylogenetic diversity index (PD whole tree)). A non-parametric permutation-based *t*-test (equivalent to the Mann–Whitney U-test) with 999 random permutations was made to assess whether the samples belonging to one experimental class were statistically different from those of a different class.

Beta-diversity analysis was performed according to the unweighted and weighted UniFrac distances among samples and represented via a Principal Coordinate Analysis (PCoA). A statistical test (i.e., the “Adonis” test, Permutational Multivariate Analysis of Variance Using Distance Matrices, using pseudo-F ratios) was used in order to define whether there was a significant difference among the experimental groups using 999 random permutations. Statistical analyses and graphs were performed in Matlab (v. 2008a, Natick, MA, USA).

#### 2.5.3. Co-Abundance Analysis

This analysis was aimed at identifying groups of bacterial genera whose abundance was correlated with each other. Spearman’s rank correlation was calculated for all the bacterial genera having abundance > 0.5% in at least 50% of the samples in each experimental category: cecum NM, cecum GF, jejunum NM, and jejunum GF. On the basis of the correlations, bacterial genera were clustered in co-abundance groups (CAGs), following the procedure originally developed by Claesson and co-workers [37] and using Euclidean distance and average linkage; Cytoscape v. 3.0 [38] was used to graphically represent CAGs, as well as the relative abundance of bacterial genera and strength of correlation.

#### 2.5.4. Microbiota-Body Weight Correlations

Spearman’s rank correlation was used to analyze the correlation between bacterial genera, the body weight, and the weight gain of the animals measured at 12 weeks of age.

#### 2.5.5. Microbial Function Prediction

Metabolic functional capacities of gut bacteria were predicted from 16S rRNA data using the Tax4Fun R package [39] and the Kyoto Encyclopedia of Genes and Genomes (KEGG). Tax4Fun transformed the SILVA-classified zOTUs into prokaryotic KEGG organisms and normalized them according to the 16S rRNA copy number. The output tables provided KEGG orthology (KO) numbers for gene annotations and Enzyme Commission (EC) numbers.

### 2.6. Statistical Analysis

The individual animal was considered the experimental unit, and data on body weight, weight gain, and blood biochemical and histological parameters were represented as means with standard deviations (SEM) by using SAS 2002 software’s GLM technique (SAS, Cary, NC, USA). A one-way ANOVA was used to compare the means, and the differences were considered significant at *p* < 0.05.

## 3. Results

### 3.1. Body Weight and Weight Gain

The mean body weight at 5 and 12 weeks of age, as well as the daily weight gain, are presented in Table 2. The body weight at 12 weeks of age and daily weight gain were greater in GF than those in the NM breed. The daily weight gain was significantly (*p* < 0.05) higher in GF rabbits than in NM rabbits (*p* = 0.048).

### 3.2. Blood Biochemical Parameters

The analysis of blood biochemical parameters in GF and NM rabbits is presented in Figure 1. Total Protein (g/dL) and albumin (g/dL) were significantly higher in GF than in NM rabbits (*p* = 0.02 and 0.034, respectively).

The comparison among the histological morphometric parameters between NM and GF rabbits in the jejunum and cecum is reported in Figure 2. In cecum, the villus width was higher in the NM group compared to the GF group (*p* = 0.001), while in the jejunum, the GF rabbits had a higher villus height with respect to the NM group (*p* ≤ 0.001).

### 3.3. Electron Microscopic Examination

The photographs of the scanning electron microscope (SEM) showed that villi on the epithelial cecum in the rabbit had a semi-zigzag pattern (Figure 3, photographs A–D). The villi on the epithelial cecum of NM showed a lower density (Photograph A) than those in GF (Photograph B). The villi tips were round and smoothed in the GF breed (Photograph D), while they were marked by irregular edges in the NM breed (Photograph C).

SEM examination of the villi on the jejunal epithelium in the rabbit revealed a tongue-like shape (Photographs E and F). The jejunal villi of NM and GF were well arranged in their dense borders, but without neat borders in NM (Photograph E), whereas in GF there were clear and neat borders (Photograph F).

### 3.4. Microbial Profile

The overall composition of the microbiota belonged to kingdom bacteria for 99.97%, with a residual of 0.03% Archaea. The main bacterial phyla in all experimental conditions included Firmicutes (average relative abundance: 58.04%), *Proteobacteria* (13.88%), *Patescibacteria* (6.54%), *Bacteroidetes* (4.64%), and *Verrucomicrobia* (3.56%), while the unclassified bacteria were 1.47% on average. With regard to bacterial families, the most abundant were: *Ruminococcaceae* (23.76%), *Enterobacteriaceae* (10.71%), *Eubacteriaceae* (8.62%), Saccharimonadaceae (6.54%), *Lactobacillaceae* (6.35%), Enterococcaceae (6.09%), *Lachnospiraceae* (4.26%), and Akkermansiaceae (3.56%). Finally, the main bacterial genera found in the samples were: *Escherichia-Shigella* (9.26%), uncultured Eubacteriaceae (8.60%), *Ruminococcaceae NK4A214 group* (6.66%), *Candidatus Saccharimonas* (6.54%), *Lactobacillus* (6.35%), *Enterococcus* (6.08%), *Ruminococcaceae UCG-014* (5.66%), *Akkermansia* (3.56%), *Christensenellaceae R-7 group* (3.10%), and *Subdoligranulum* (3.08%). About 37% of the average relative abundance was due to genera whose rel. ab. was <3% on average.

Alpha-diversity estimations for cecum and jejunum samples in both rabbit breeds (NM and GF) showed that the diversity across the different experimental conditions was comparable for all the metrics analyzed (with the only exception of a significant difference between jejunum and cecum samples of the GF breed, *p* = 0.011), as shown in Figure 4. For beta-diversity analysis, a significant difference was found for the intestine tract experimental variable, with cecum and jejunum samples being different (regardless of the rabbit breed) for both the unweighted and the weighted UniFrac distances (*p* = 0.022 and *p* = 0.001, respectively, data not shown). Stratifying by organ and breed, a significant difference was found between cecum and jejunum samples of the GF breed. No difference was found for the NM breed when comparing the two breeds for jejunum and cecum samples separately (Figure 5), this statement is right

### 3.5. Comparison between Breeds in Cecum Samples

For microbiota composition, the most abundant phyla were Firmicutes, with an average relative abundance of 54.0% and 65.4% in NM and GF breeds, respectively, followed by Patescibacteria (7.3% and 10.3%) and Bacteroidetes (7.2% and 3.5%). The average relative abundances for genera having an average abundance > 1% in the cecum of NM and GF breeds are reported in Table 3. At the genus level, the *p*-value indicated no significant difference (*p* < 0.05) for any genera between GF and NM breeds.

### 3.6. Comparison between Breeds in Jejunum Samples

In jejunum samples, a high relative abundance at phylum level was found for *Firmicutes* (44.0% and 40.2% in NM and GF breeds, respectively), followed by *Proteobacteria* (19.6% and 24.2%). The average relative abundances for main genera (average abundance > 1%) in the jejunum of NM and GF breeds are shown in Table 4. At genus level, no different significances (*p* < 0.05) between the GF and NM breeds were found. However, GF showed a tendency towards an increase in Bacteroides and towards a depletion of unclassified members of the *Eubacteriaceae* family and the *Christensenellaceae R-7 group*.

### 3.7. Comparisons between Organs in NM Breed

The phylum-level composition of the microbiota in cecum and jejunum samples from the NM breed was dominated by *Firmicutes* (average rel ab.: 52.9% and 45.8%, respectively), followed by *Proteobacteria* (0.1% vs. 18.9%), and *Bacteroidetes* (7.2% vs. 2.4%). Seven out of the 17 identified genera in the *Firmicutes* phylum were significantly (*p* < 0.05) different between the examined organs. Among them, bacteria from the *Lactobacillales* order were more abundant in jejunum samples than in cecum (avg. rel. ab.: 23.0% vs. 9.9%) and, in particular, the *Enterococcus* genus. On the other hand, *uncultured Clostridiales vadinBB60 group*, *Ruminococcaceae NK4A214 group*, *Ruminiclostridium 5*, and members of the family *Ruminococcaceae* (all belonging to the *Firmicutes* phylum) were significantly (*p* < 0.05) higher in the cecum than the jejunum (Table 5). The *Escherichia-Shigella* genus (phylum *Proteobacteria*) was also significantly higher in jejunum samples (Table 5).

### 3.8. Comparisons between Samples in GF Breed

In the GF rabbit breed, at the phylum level, the average relative abundance of Firmicutes was higher in the cecum than the jejunum (61.8% vs. 41.3%, respectively), while Proteobacteria were present in the jejunum (average rel. ab.: 24.2%) and nearly absent in cecum (average rel. ab.: 0.8%), as shown in Table 6. At the genus level, 12 out of 16 genera identified in the *Firmicutes* phylum were significantly (*p* < 0.05) different between organ samples. In particular, nine genera (i.e., *Bacillus*, *Lactobacillus*, *Christensenellaceae R-7 group*, *Lachnospiraceae (other)*, *Ruminococcaceae NK4A214 group*, *Ruminococcaceae UCG-013*, *Ruminococcaceae UCG-014*, *Ruminococcus 2*, and *Subdoligranulum*) were significantly higher in the cecum than the jejunum of the GF breed (Table 6). On the other hand, *Enterococcus, Sarcina*, and *Dubosiella* genus were higher in the jejunum than the cecum (Table 6).

Regarding bacteria belonging to other phyla, the genus *Candidatus Saccharimonas* (phylum: *Patescibacteria*) was significantly higher in the cecum, whereas the *Escherichia-Shigella* genus (phylum: *Proteobacteria*) was significantly higher in the jejunum (Table 6).

### 3.9. Correlation and Co-Abundance Analysis

Co-abundance analysis aimed to infer interactions between bacteria genera in the gut microbiota of NM and GF breeds and organ sampling places in the cecum and jejunum. Clustering the matrix of the pairwise Spearman’s correlations between the main bacterial genera in the experiment highlighted four main co-abundant groups (CAGs), i.e., genera whose abundance was concordantly increased/decreased throughout all the 40 samples. CAG1 included genera from the families of *Ruminococcaceae* (*Ruminococcus 5, Ruminococcus 1*, *other Ruminococcus, Ruminococcaceae UCG-014, Ruminococcaceae UCG-013*, *and Ruminococcaceae NK4A214*), and *Lachnospiraceae* (*unclassified Lachnospiraceae, Lachnospiraceae NK4A136*)*,* plus other genera such as *Akkemansia*, *Subdoligranulum*, *Christenellaceae R-7 group*, *uncultured Gastroaerophilales* and *Candidatus Saccharimonas*, all highly correlated one to each other; CAG2 comprised the uncultured members of the *Eubacteriaceae* family and some unclassified *Atopobiaceae*; CAG3 was composed by bacteria unclassified at lower phylogenetic levels; the fourth (CAG4) was made only by Escherichia-Shigella genus, which was correlated negatively with CAGs1-3 (Appendix A).

The samples from NM breed cecum were characterized by the involvement of Atopobiaceae and uncultured members of *Eubacteriaceae* in the correlations with the main cluster of genera and by relatively few connections between *Lachnospiraceae NK4A136*, *Ruminococcus 1*, and *Lachnospiraceae* (Figure 6). In cecum samples from GF breeds, the analyzed bacterial groups had few correlations and seemed to be more independent of each other (Figure 6). In cecum samples, regardless of the rabbit breed, Escherichia-Shigella (even at low abundance) was inversely correlated to the other genera.

In jejunum samples, the network of correlations among the genera of the main cluster involves many or quite all the bacteria. Even though *Escherichia-Shigella* was relatively abundant, it was uncorrelated with the other bacteria. In the jejunum samples from the NM breed, a correlation was evident also between the members of the *Atopobiaceae* and *Eubacteriaceae* families and other unclassified bacteria (Figure 6). In the jejunum of GF breed samples, these further correlations were not significant, and the network involved only the main cluster of bacteria (Figure 6).

### 3.10. Microbiota-Body Weight Correlations

Figure 7 reports the correlations estimated between the twenty most abundant genera in the gut microbiota, the body weight at 12 weeks, and the weight gain, separated for the two organs and the two breeds. In cecum samples from the NM breed, *Escherichia-Shigella* (relative abundance: 0.8%) was significantly (*p* < 0.05) correlated with the weight gain.

In GF cecum samples, the abundances of the Lactobacillus genus (average 8.3%) were significantly (*p* < 0.05) correlated with both the weight at 12 weeks and the weight gain. On the other hand, in the jejunum of NM samples, a significant correlation was found between the *Enterococcus* abundances (average: 13.7%) and both the weight at 12 weeks and the weight gain.

### 3.11. Microbial Function Prediction

To explore the effects of host genetics on the potential functional capacities of the gut microbiome, functional profiles of cecum and jejunum bacterial communities in NM and GF breeds were predicted based on 16S rRNA sequencing data. Predictions of the functional capacities of gut bacteria were based on the Kyoto Encyclopaedia of Genes and Genomes (KEGG) pathways, identifying a total of 6419 KEGG Orthologies (KOs) in all samples (Appendix A). Functional predictions in the gut microbiome belonged to signaling and cellular processes (secretion system and transporters), genetic information processing (chaperones and folding catalysts, transcription, transfer RNA biogenesis, and translation), cellular processes (growth factor and mobility), and metabolism (amino acids, carbohydrates, lipids, cofactors, and vitamins) categories, as presented in Figure 8A–D, respectively. The functional predictions were not significantly different between NM and GF breeds in both jejunum and cecum except for the growth factor and mobility pathway (cellular processes) of the jejunum samples (Figure 8C), which resulted in higher levels in the GF breed than the NM breed. On the other hand, significant differences were reported for the jejunum vs. cecum comparison: both NM and GF breeds showed differences in transporters, transcription, carbohydrate metabolism, and lipid metabolism; the GF breed was also characterized by a differential abundance in the pathways of the secretion system, chaperones and folding catalysts, translation, and metabolism of cofactors and vitamins; and finally, NM breed samples showed a significant difference in the pathway of amino acid metabolism. In all comparisons, the functional potential of cecum samples was higher than that of jejunum samples.

## 4. Discussion

The body weight and weight values in NM and GF rabbits were similar to the findings (NM, 1169.4 vs. GF, 1327.7) of our previous work [40]. Moreover, we already reported that GF rabbits were significantly heavier and had a higher relative growth rate as compared to NM rabbits. NM is a rabbit breed established in three governorates in Middle Egypt by the Animal Production Research Institute (APRI) [41] and, according to the general classification used in the European Rabbit Breed Standards Book [42], belongs to the small-sized breed, while GF belongs to the medium-sized breed.

Total proteins and globulins were significantly affected by the breed, as documented in previous studies in four breeds [43] and three breeds [44] under Egyptian environmental conditions. High globulin concentrations in GF rabbits could be attributed to increasing the gamma globulin fraction, which indicates immunity status [45]. Previous studies have indicated that an altered serum component profile could reflect differences in the gut microbiome of rabbits [31], which is consistent with our findings as shown in Figure 4. This study suggested that host rabbit breeds can shape the gut microbiome and serum metabolome, including the interactions among the host-gut microbiome and serum metabolome that are important. Thus, total proteins and globulins may be indicators of the health status and production traits of meat rabbits.

The villi on the jejunum and cecum were higher in GF than NM rabbits in addition, the villi of the cecum in MN were less dense and their tips were marked by irregular edges, as shown in the SEM examination. These observations may be due to some kind of toxins produced in the intestine and cecum [46] by bacteria that were in close contact only with epithelial cells that had lost their brush border [47]. In addition, the fermentation operated by bacteria could produce hydrogen ions that can induce damage in the cecum and jejunum mucosa [46].

The microbiota composition of the samples was not significantly different when comparing the two rabbit breeds (NM and GF) within the same intestinal tract. In fact, among the major constituents of the microbiota (relative abundances > 1%), none of them were found to be significantly diverse. On the other hand, differences were more evident when comparing the jejunum vs. cecum microbiota, in particular in the GF breed, where both alpha- and beta-diversity estimations were found to be significantly different. This was also reflected when considering the functional predictions deriving from the microbial profiles, with jejunum and cecal samples appearing very different from one another. The predominant phylum in microbiota samples for both intestinal tracts (jejunum and caecum) was *Firmicutes*, followed by *Patescibacteria* and *Bacteroidetes* (in cecum) and *Proteobacteria* and *Actinobacteria* (in jejunum), in accordance with the fact that *Firmicutes* was found to be the most dominant phylum in the rabbit microbiota, regardless of source, age, and season [14]. Our results are also in harmony with the findings of Fu et al. [48], who reported that Firmicutes was the most dominant phylum in the foregut and hindgut of rabbits, while the second most dominant one was Proteobacteria (in the foregut) or Bacteroidetes (in the hindgut). Another study reported that Firmicutes were the most abundant phylum in all of the sections of the gastro-intestinal tract examined (45.9%), such as the stomach, duodenum, jejunum, ileum, cecum, and colon, followed by *Bacteroidetes* in the large intestine (38.9%), *Euryarchaeota* (29.6%), and *Patescibacteria* (13.8%) in the foregut, especially in jejunum [13]. The abundance of members of the Firmicutes phylum was higher in GF than NM rabbits, both in the cecum (73.5% GF vs. 63.9% NM) and jejunum (44.0% GF vs. 40.2% NM) samples. Previous studies have shown that a high abundance of *Firmicutes*, as shown in GF, can enhance intestinal mucosa and reduce oxidative stress in the intestinal tract in piglets [49,50]. This concept was further supported by histological examinations and scanning electron microscopy (SEM) in the cecum and jejunum villi of GF rabbits.

Within the *Firmicutes* phylum, the most abundant order was *Clostridiales* and, among genera, those from the *Ruminococcaceae* and the *Lachnospiraceae* families. As highlighted in the co-abundance groups’ analysis, the abundance of the members of these two families was highly correlated, establishing a sort of “core” microbiota. A high abundance of *Clostridiales*, *uncultured Clostridiales vadinBB60 group*, and *Ruminiclostridium 5*, such as that highlighted in the jejunum of GF, may reduce the effect of *Escherichia coli* as a pathogenic agent through remodeling the signaling pathway [51]. The high abundance of *Bacteroides* and *Ruminococcus* in the cecum of GF breed rabbits could be related to a healthy gut, in accordance with the findings of previous studies on rabbits [31].

In cecum samples, members of the *Eubacteriaceae* family occupied a central position in the interaction network, with positive correlations observed between *Lachnospiraceae*, *Ruminiclostridium*, and *Eubacteriaceae*, possibly due to their functional potential in the specific metabolic pathways. *Eubacteriaceae* and *Lachnospiraceae* bacteria exhibited metabolic specificities for pyruvate and carbohydrate degradation [24]. Members of *Ruminococcaceae* seemed to be highly specialized in pyruvate-to-lactate fermentation [52,53], as happens in the degradation of plant material such as pectin and cellulose in the colonic fermentation of dietary fibers in mammals [54,55]. These results could be confirmed by Ye et al. [31], who reported that families *Ruminococcus* and *Lachnospiraceae* could be considered biomarkers for improving the health and production performance of meat rabbits. Moreover, other low-abundance microorganisms, such as the *Akkermansia* genus (*Verrucomicrobiales* order), could play a key role in the hydrolysis of diverse ingested polysaccharides and contribute to a more complete digestion of dietary cellulose [51,56]. The significant decrease in the proportion of *Verrucomicrobia* in the jejunum of the NM breed as compared to the cecum suggests less optimal jejunum health as well as a more pro-inflammatory state, as already reported in mice [57]. The positive correlations among *Ruminiclostridium*, *Lachnospiraceae*, and *Eubacteriaceae* that were observed in the cecum with increasing *Verrucomicrobia* in the jejunum of the GF could explain the significant difference in the microbiota composition between organs in the GF breed.

The abundance of the *Escherichia-Shigella* genus in the cecum of NM breed rabbits was positively correlated to body weight gain, as were Enterococcus and both body weight at 12 weeks and body weight gain in the jejunum of the same animals. *Escherichia-Shigella* should be considered a pathogen in hosts [9]; meanwhile, *Enterococcus* could be considered a beneficial bacteria that could produce bacteriocins active against bacteria such as Listeria and indigenous clostridia in the gut of rabbits [58]. On the other hand, in the cecum of the GF rabbits, there were significant correlations between Lactobacillus and both the body weight at 12 weeks and the body weight gain. *Lactobacillus* can promote the fermentation of carbohydrates into lactic acid and intestinal health [51], increase the concentration of short-chain fatty acids (SCFAs) in the intestines of mice, promote the growth of intestinal epithelial cells [59], and have a biological antagonistic effect on pathogenic bacteria such as *E. coli*. These results confirmed that the cecum is the main organ harboring the microbial fermentation processes in the gut of the rabbit [16], and hosting *Lactobacillus*, as occurred in the GF breed, could cause increased growth.

The functional potential of cecum samples was higher than that of jejunum samples, in particular for the secretion system, transporters, amino acids and carbohydrate metabolism, ribosome biogenesis, and translation. This could be attributed to the fact that cecum is the richest and most diverse microbial community in the rabbit gut [14,60]. The significantly different carbohydrates and lipid metabolic activity between the cecum and jejunum we observed could be due to the difference in microbial communities and their capacity to ferment to obtain metabolic energy [9].

The limited number of replicates per condition (n = 10) and the high variability observed in the microbial profiles of the samples could have negatively influenced a better characterization of the differences in the microbiota composition inherent to the two breeds. On the other hand, we were able to highlight the different bacterial communities inhabiting the jejunum and cecum tracts of both rabbit breeds and gain insight into the specific functions they preside over.

## 5. Conclusions

This study characterized the composition of the rabbit jejunal and cecal microbiota as well as their potential influence on rabbit growth. Our study provides that *Patescibacetria* and *Bacteroides* were major constituents of the microbiota in the rabbit hindgut, as well as *Proteobacteria* (mostly made up by members of the *Escherichia-Shigella* genus) in the foregut. Firmicutes was the most abundant phylum in both intestinal tracts. Within the Firmicutes phylum, the most abundant genera were all members of the *Ruminococcaceae* and *Lachnospiraceae* families, which are generally beneficial key members of the gut ecosystem, have multiple interactions with the other members of the gut microbiota, and may also reduce the pathogenetic effect of *Escherichia coli*. The high abundance of *Bacteroides* and *Ruminococcus* in the cecum of GF-bred rabbits could indicate a more healthy gut, as shown by the rounding and smoothing of the villi tips and the high density in the epithelium, which reflected in a high growth rate. A better understanding of the relationship between gut microbiota and the factors influencing its composition could improve the management and health of the rabbit.

## Figures and Tables

**Figure 1 animals-13-02294-f001:**
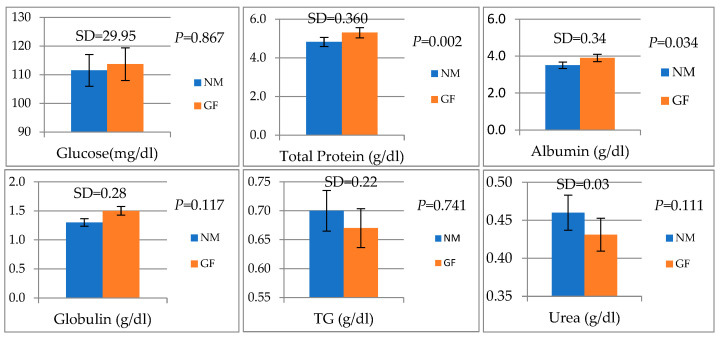
Blood biochemical parameters in GF and NM rabbits. Means and standard deviations (SD), and the differences were considered significant at *p* < 0.05.

**Figure 2 animals-13-02294-f002:**
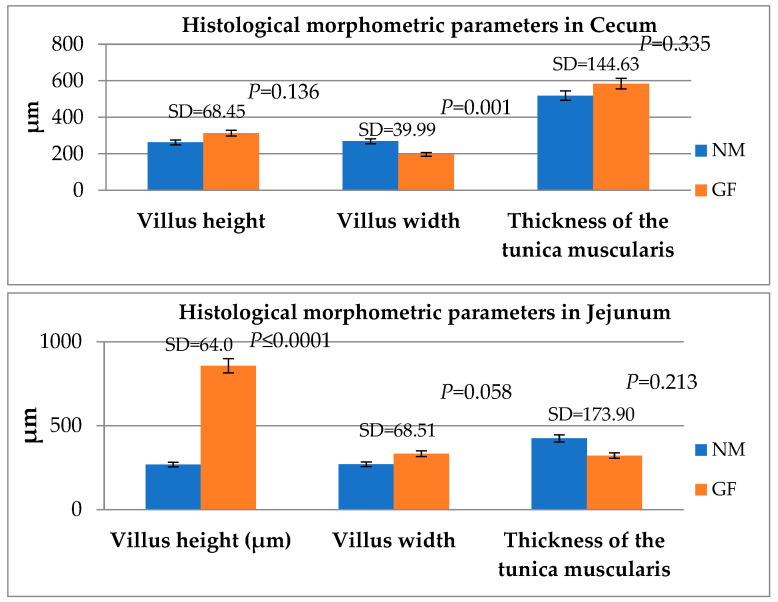
Comparing the histological morphometric parameters between GF and NM rabbits in the cecum and jejunum. Means and standard deviations (SD), and the differences were considered significant at *p* < 0.05.

**Figure 3 animals-13-02294-f003:**
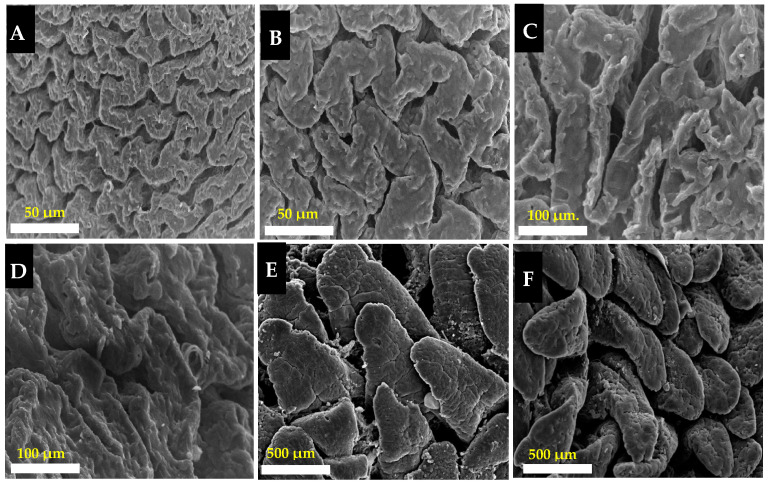
SEM photographs of villi of the cecum and jejunum in NM and GF breeds. (**A**,**C**) photographs showed the villi in the cecum of NM. (**B**,**D**) photographs showed the villi on the cecum of the GF breed. (**E**,**F**) photographs of the villi on the jejunum of the NM and GF breeds, respectively. Scale bars: (**A**,**B**) 50 µm; (**C**,**D**) 100 µm; (**E**,**F**) 500 µm.

**Figure 4 animals-13-02294-f004:**
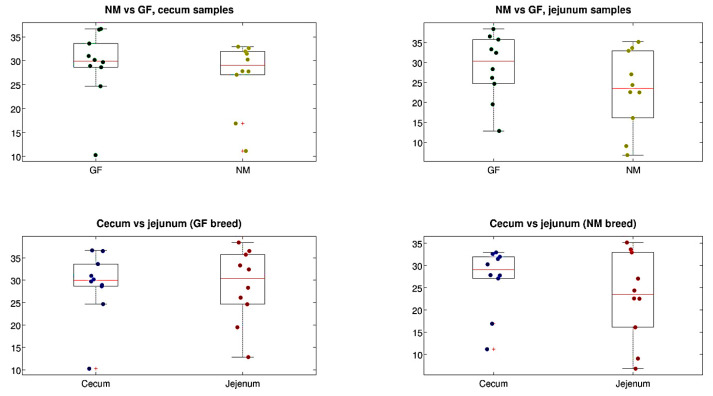
Boxplots of alpha-diversity estimations (PD whole tree metric) for rabbit breeds and organs separated. Each point represents a sample, whereas the median value is in the red line.

**Figure 5 animals-13-02294-f005:**
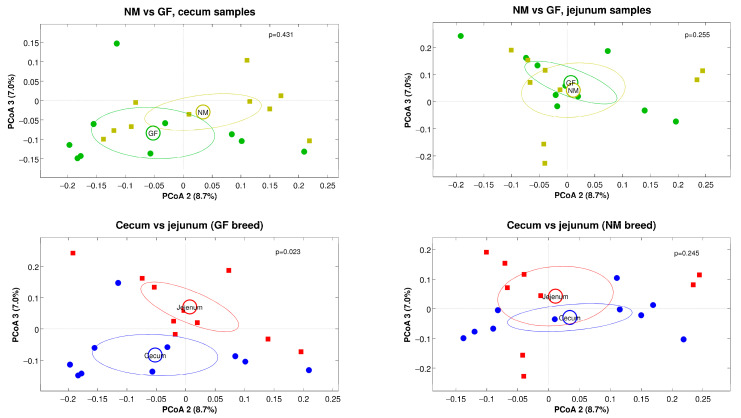
Principal Coordinate Analysis (PCoA) plots derived from unweighted UniFrac distances comparing the two breeds and organs separately. In the plots, each point represents a sample, colored according to the experimental group. Ellipses represent the 95% SEM-based confidence, and the *p*-values reported are those deriving from the “adonis” test on the distance matrices. The second and third principal coordinates are plotted.

**Figure 6 animals-13-02294-f006:**
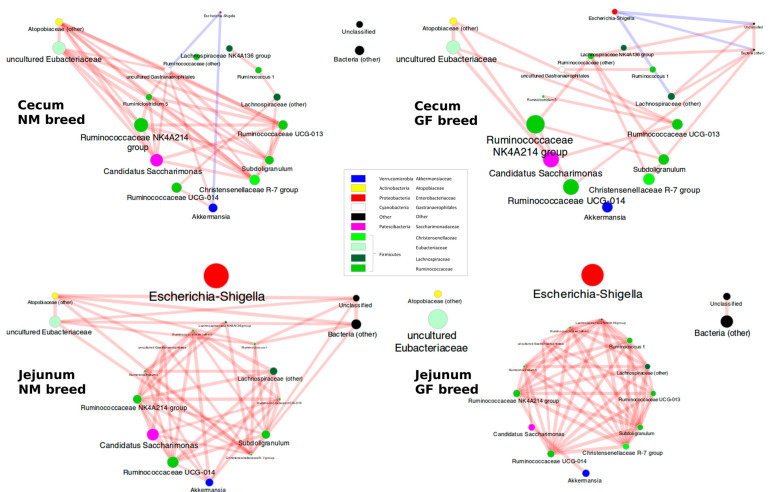
The Co-Occurrence patterns among the bacterial genera in the cecum and jejunum in NM and GF breeds, separately. Patterns here depict the significant (*p* < 0.05) correlations among the bacterial genera for the four experimental classes. Node size is proportional to the average abundance of the genera in the experimental condition, and the fill color follows the taxonomic classification of the genera. The edges size is proportional to the strength of correlation, whereas the edges color is blue for negative correlations and red for positive correlations.

**Figure 7 animals-13-02294-f007:**
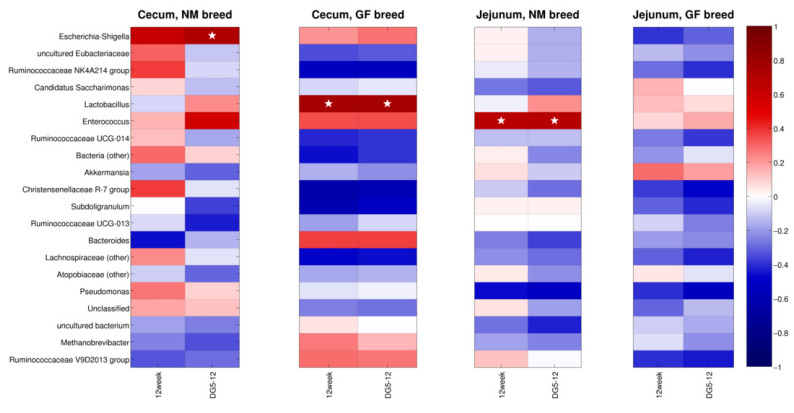
Heatmap of the Spearman’s correlation between the top twenty most abundant genera, the body weight, and the weight gain in the experimental classes. Blue is for negative correlations, red for positive correlations, and “★” indicates significant (*p* < 0.05) correlations.

**Figure 8 animals-13-02294-f008:**
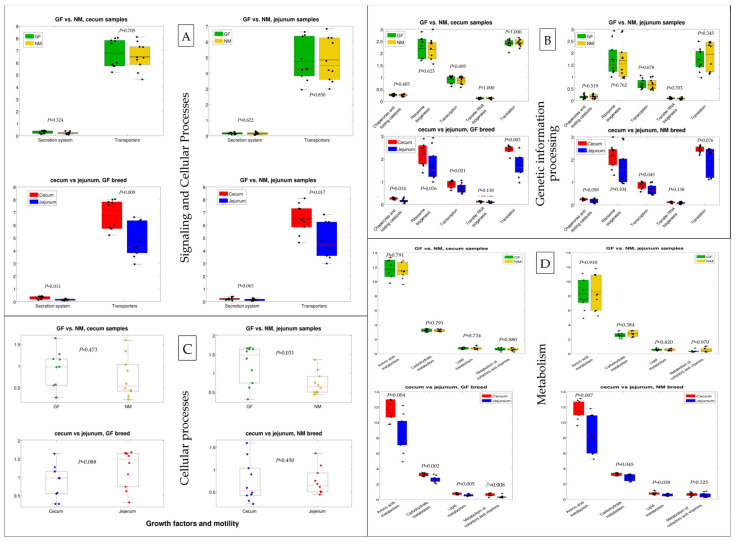
The functional prediction of gut microbiome in rabbit breeds NM and GF, separated by 0.008 organs (cecum and jejunum). (**A**) Signaling and cellular processes; (**B**) Genetic information processing; (**C**) Cellular processes; (**D**) Metabolism.

**Table 1 animals-13-02294-t001:** Ingredients and chemical composition of the diet.

Ingredient	%
Barseem	31.7
Barley	18.0
Corn	10.0
Soybean meal 44%	11.0
Wheat bran	27.0
Limestone	1.0
Di-Calcuim	0.5
Methionine	0.2
Salt	0.3
Premix	0.3
Total	100
Chemical Analysis:	
Crude protein%	16.4
Crude fiber%	12.7
Digestible energy (Kcal/Kg)	2403

**Table 2 animals-13-02294-t002:** Body weight and weight gain in GF and NM rabbits.

	Breed	NM	GF	SEM	*p* Value
Items					
Body Weigh at Week5		542.5	490.5	30.9	0.249
Body Weigh at Week12		1093.0	1229.0	83.7	0.265
Daily weight gain (DWG)		11.2	15.0	1.5	0.048

**Table 3 animals-13-02294-t003:** Average of the relative abundance and standard deviation (between parentheses) at genus level for microbiota composition in the cecum of NM and GF breeds. The average abundance over the replicates (n = 10) of the same condition, as well as (between parenthesis) the standard deviation, are the results of the Mann–Whitney U-test. Only genera with an average abundance > 1% were shown. Average abundances at higher levels are calculated by summing up the genera belonging to each order and phylum.

Phylum	NM	GF	Order	NM	GF	Genus	NM	GF	*p*-Value
Firmicutes	54	65.4	Bacillales	1.0	1.7	Bacillus	1.0 (3.2)	1.7 (5.5)	0.387
			Lactobacillales	9.9	8.1	Enterococcus	1.6 (3.5)	0.5 (1)	0.808
						Lactobacillus	8.3 (23.7)	7.6 (24.1)	0.382
			Clostridiales	43.1	55.6	Christensenellaceae R-7 group	4.9 (4.1)	6.0 (4.6)	0.571
						Unclassified Clostridiales vadinBB60 group	1.1 (3.1)	0.1 (0.3)	0.679
						uncultured Clostridiales vadinBB60 group	1.3 (2.4)	0.7 (1.6)	0.149
						uncultured Eubacteriaceae	7.1 (6.5)	7.7 (5)	0.623
						Lachnospiraceae (other)	2 (1.6)	1.9 (1.4)	0.940
						Lachnospiraceae NK4A136 group	1.4 (1.7)	0.9 (0.9)	0.597
						Ruminiclostridium 5	1.2 (1.1)	0.3 (0.4)	0.082
						Ruminococcaceae (other)	1.1 (0.8)	0.8 (0.5)	0.307
						Ruminococcaceae NK4A214 group	8.5 (6.7)	12.6 (7.6)	0.273
						Ruminococcaceae UCG-013	4.1 (3.9)	5.3 (2.8)	0.733
						Ruminococcaceae UCG-014	4.7 (3.4)	10.1 (6.7)	0.064
						Ruminococcaceae V9D2013 group	0.6 (0.9)	1.9 (4.3)	1.000
						Ruminococcus 1	1.2 (2)	1.0 (1.3)	0.850
						Ruminococcus 2	0.5 (0.6)	1.2 (1.4)	0.272
						Subdoligranulum	3.4 (3.7)	5.1 (3.7)	0.212
Bacteroidetes	7.2	3.5	Bacteroidales	7.2	3.5	Bacteroides	4.5 (9.6)	0.5 (0.9)	0.256
						uncultured Muribaculaceae	1.7 (3.7)	2.1 (6.4)	0.451
						Alistipes	1 (1.4)	0.9 (1.6)	0.467
Actinobacteria	1.6	1.6	Coriobacteriales	1.6	1.6	Atopobiaceae (other)	1.6 (1.8)	1.6 (1.7)	0.850
Patescibacteria	7.3	10.3	Saccharimonadales	7.3	10.3	Candidatus Saccharimonas	7.3 (8.2)	10.3 (6.3)	0.406
Verrucomicrobia	3.7	5.3	Verrucomicrobiales	3.7	5.3	Akkermansia	3.7 (3.1)	5.3 (5.6)	0.820
Cyanobacteria	1.3	0.6	Gastranaerophilales	1.3	0.6	uncultured Gastranaerophilales	1.3 (1.3)	0.6 (0.5)	0.496

**Table 4 animals-13-02294-t004:** Average of the relative abundance and standard deviation (between parentheses) at genus level for microbiota in the jejunum for NM and GF breeds. The average abundance over the replicates (n = 10) of the same condition, as well as (between parenthesis) the standard deviation, are shown. The *p*-value refers to that of a two-tailed Mann–Whitney U-test. Only genera with an average abundance > 1% were shown. Average abundances at higher levels are calculated by summing up the genera belonging to each order and phylum.

Phylum	NM	GF	Order	NM	GF	Genus	NM	GF	*p*-Value
Firmicutes	44	40.2	Lactobacillales	23	12.8	Enterococcus	13.7 (21.5)	8.5 (21.8)	0.472
						Lactobacillus	6.4 (13.9)	3.1 (4.6)	0.496
						Weissella	2.9 (9.3)	1.2 (3.8)	0.804
			Clostridiales	20.6	24.1	Christensenellaceae R-7 group	0.2 (0.4)	1.3 (1.7)	0.088
						Sarcina	0 (0)	1.5 (3.4)	0.451
						uncultured Eubacteriaceae	5.7 (5.5)	14 (10.9)	0.054
						Lachnospiraceae (other)	2.4 (4.5)	0.6 (0.8)	0.940
						Ruminococcaceae NK4A214 group	3.3 (3.3)	2.2 (2.1)	0.545
						Ruminococcaceae UCG-014	5.8 (7.9)	2.1 (2.3)	0.762
						Ruminococcaceae V9D2013 group	0.1 (0.2)	1.7 (3.3)	0.384
						Subdoligranulum	3.1 (8.7)	0.7 (0.8)	0.596
			Erysipelotrichales	0.4	3.3	Dubosiella	0.2 (0.3)	1.9 (2.8)	0.116
						Erysipelotrichaceae (other)	0.2 (0.3)	1.4 (2.3)	0.180
Proteobacteria	19.6	24.2	Enterobacteriales	19	19.1	Citrobacter	0.1 (0.1)	1.8 (5.3)	0.539
						Escherichia-Shigella	18.9 (26.6)	17.3 (23)	0.571
			Pseudomonadales	0.6	5.1	Pseudomonas	0.6 (0.6)	5.1 (14.2)	0.791
Actinobacteria	1.2	3.2	Coriobacteriales	0.7	2.2	Atopobiaceae (other)	0.7 (0.8)	2.2 (3.4)	0.450
			Coriobacteriales	0.5	1	Eggerthellaceae (other)	0.5 (0.7)	1 (1.1)	0.344
Bacteroidetes	2.1	0.4	Bacteroidales	2.1	0.4	Bacteroides	2.1 (5.1)	0.4 (0.4)	0.064
Verrucomicrobia	2.7	2.5	Verrucomicrobiales	2.7	2.5	Akkermansia	2.7 (5.3)	2.5 (6.5)	0.910
Patescibacteria	6.6	1.9	Saccharimonadales	6.6	1.9	Candidatus Saccharimonas	6.6 (9.7)	1.9 (1.8)	0.940

**Table 5 animals-13-02294-t005:** Average relative abundances and the standard deviation (between parentheses) at genus level for microbiota in the cecum and jejunum in the NM breed. The average abundance over the replicates (n = 10) of the same condition, as well as (between parenthesis) the standard deviation, are shown. The *p*-value refers to that of a two-tailed Mann–Whitney U-test. “*” indicated statistical significance (*p* < 0.05). Only genera with an average abundance > 1% were shown. Average abundances at higher levels are calculated by summing up the genera belonging to each order and phylum.

Phylum	Cecum	Jejunum	Order	Cecum	Jejunum	Genus	Cecum	Jejunum	*p*-Value
Firmicutes	52.9	45.8	Bacillales	1	0.4	Bacillus	1.0 (3.2)	0.4 (0.7)	0.146
			Lactobacillales	9.9	23	Enterococcus	1.6 (3.5)	13.7 (21.5)	0.022 *
						Lactobacillus	8.3 (23.7)	6.4 (13.9)	0.405
						Weissella	0.0 (0.0)	2.9 (9.3)	0.871
			Clostridiales	42	22.4	Christensenellaceae R-7 group	4.9 (4.1)	0.2 (0.4)	0.013
						Unclassified Clostridiales vadinBB60 group	1.1 (3.1)	0.3 (0.7)	0.936
						uncultured Clostridiales vadinBB60 group	1.3 (2.4)	0.2 (0.5)	0.036 *
						uncultured Eubacteriaceae	7.1 (6.5)	5.7 (5.5)	0.597
						Lachnospiraceae (other)	2.0 (1.6)	2.4 (4.5)	0.173
						Lachnospiraceae NK4A136 group	1.4 (1.7)	0.1 (0.1)	0.009 *
						Ruminiclostridium 5	1.2 (1.1)	0.4 (0.8)	0.031 *
						Ruminococcaceae (other)	1.1 (0.8)	0.3 (0.4)	0.017 *
						Ruminococcaceae NK4A214 group	8.5 (6.7)	3.3 (3.3)	0.076
						Ruminococcaceae UCG-013	4.1 (3.9)	0.4 (0.4)	0.075
						Ruminococcaceae UCG-014	4.7 (3.4)	5.8 (7.9)	0.450
						Ruminococcus 1	1.2 (2.0)	0.2 (0.4)	0.058
						Subdoligranulum	3.4 (3.7)	3.1 (8.7)	0.096
Bacteroidetes	7.2	2.4	Bacteroidales	7.2	2.4	Bacteroides	4.5 (9.6)	2.1 (5.1)	0.240
						uncultured Muribaculaceae	1.7 (3.7)	0.1 (0.1)	0.721
						Alistipes	1.0 (1.4)	0.2 (0.4)	0.050 *
Proteobacteria	0.1	18.9	Enterobacteriales	0.1	18.9	Escherichia-Shigella	0.1 (0.3)	18.9 (26.6)	0.011 *
Patescibacteria	7.3	6.6	Saccharimonadales	7.3	6.6	Candidatus Saccharimonas	7.3 (8.2)	6.6 (9.7)	0.880
Verrucomicrobia	3.7	2.7	Verrucomicrobiales	3.7	2.7	Akkermansia	3.7 (3.1)	2.7 (5.3)	0.307
Actinobacteria	1.6	0.7	Coriobacteriales	1.6	0.7	Atopobiaceae (other)	1.6 (1.8)	0.7 (0.8)	0.472
Cyanobacteria	1.3	0.3	Gastranaerophilales	1.3	0.3	uncultured Gastranaerophilales	1.3 (1.3)	0.3 (0.4)	0.103

**Table 6 animals-13-02294-t006:** Average relative abundances and the standard deviation (between parentheses) at genus level for microbiota in the cecum and jejunum in the GF breed. The average abundance over the replicates (n = 10) of the same condition, as well as (between parenthesis) the standard deviation, are shown. The *p*-value refers to that of a two-tailed Mann–Whitney U-test. “*” indicated statistical significance (*p* < 0.05). Only genera with an average abundance > 1% were shown. Average abundances at higher levels are calculated by summing up the genera belonging to each order and phylum.

Phylum	Cecum	Jejunum	Order	Cecum	Jejunum	Genus	Cecum	Jejunum	*p*-Value
Firmicutes	61.8	41.3	Bacillales	1.7	0.5	Bacillus	1.7 (5.5)	0.5 (1.4)	0.002 *
			Lactobacillales	8.1	12.8	Enterococcus	0.5 (0.01)	8.5 (21.8)	0.032 *
						Lactobacillus	7.6 (24.1)	3.1 (4.6)	0.010 *
						Weissella	0.0 (0.0)	1.2 (3.8)	0.754
			Clostridiales	51.8	24.7	Christensenellaceae R-7 group	6.0 (4.6)	1.3 (1.7)	0.011 *
						Sarcina	0.0 (0.0)	1.5 (3.4)	0.035 *
						uncultured Eubacteriaceae	7.7 (5.0)	14 (10.9)	0.212
						Lachnospiraceae (other)	1.9 (1.4)	0.6 (0.8)	0.026 *
						Ruminococcaceae NK4A214 group	12.6 (7.6)	2.2 (2.1)	0.005 *
						Ruminococcaceae UCG-013	5.3 (2.8)	0.5 (0.5)	0.002 *
						Ruminococcaceae UCG-014	10.1 (6.7)	2.1 (2.3)	0.009 *
						Ruminococcaceae V9D2013 group	1.9 (4.3)	1.7 (3.3)	0.940
						Ruminococcus 2	1.2 (1.4)	0.1 (0.1)	0.031 *
						Subdoligranulum	5.1 (3.7)	0.7 (0.8)	0.006 *
			Erysipelotrichales	0.2	3.3	Dubosiella	0.1 (0.1)	1.9 (2.8)	0.031 *
						Erysipelotrichaceae (other)	0.1 (0.2)	1.4 (2.3)	0.074
Patescibacteria	10.3	1.9	Saccharimonadales	10.3	1.9	Candidatus Saccharimonas	10.3 (6.3)	1.9 (1.8)	0.004 *
Proteobacteria	0.8	24.2	Enterobacteriales	0.8	24.2	Citrobacter	0.0 (0.0)	1.8 (5.3)	0.571
						Escherichia-Shigella	0.7 (1.6)	17.3 (23)	0.007 *
						Pseudomonas	0.1 (0.3)	5.1 (14.2)	0.074
Actinobacteria	2.2	3.2	Coriobacteriales	2.2	3.2	Atopobiaceae (other)	1.6 (1.7)	2.2 (3.4)	0.821
						Eggerthellaceae (other)	0.6 (0.5)	1 (1.1)	0.597
Verrucomicrobia	5.3	2.5	Verrucomicrobiales	5.3	2.5	Akkermansia	5.3 (5.6)	2.5 (6.5)	0.021 *
Bacteroidetes	2.1	0.9	Bacteroidales	2.1	0.9	uncultured Muribaculaceae	2.1 (6.4)	0.9 (2.8)	0.470

## Data Availability

All raw data from microbial genome sequencing have been uploaded to the National Center for Biotechnology Information (NCBI) and can be found under BioProject ID: PRJNA992887 (https://www.ncbi.nlm.nih.gov/bioproject/PRJNA992887, accessed on 3 July 2023).

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
