# Peer review of "Diversity and Co-Occurrence Pattern Analysis of Cecal and Jejunal Microbiota in Two Rabbit Breeds"

_animals, 2023, doi:10.3390/ani13142294_

Round 1
Reviewer 1 Report
The manuscript presents data comparing and contrasting the microbiomes of two distinct rabbit species fed similar diets until the age of 12 weeks when sampling was done.
The data is quite interesting, showing that perhaps villus health may be associated with certain bacterial families in the gut and may, subsequently, affect weight gain and final body weights.
The science is sound, experimental design and statistical analysis and conclusions made are appropriate.
The reviewer has one comment:
Since there were numerous rabbits from each breed originally included in the study, could the authors not have sampled at more timepoints during the experiment?
There are also minor English/Grammar corrections needed.
These are very minor.
Author Response
Dear reviewer 1
Greetings for the day!
Manuscript ID: animals- 2399147 “Title: Diversity and Co-occurrence Pattern Analysis of Cecal and Jejunal Microbiota in Two Rabbits Breeds”
Thank you and all the reviewers for pointing out those details.
Please below that we respond to your comments point to point.
The manuscript presents data comparing and contrasting the microbiomes of two distinct rabbit species fed similar diets until the age of 12 weeks when sampling was done.
The data is quite interesting, showing that perhaps villus health may be associated with certain bacterial families in the gut and may, subsequently, affect weight gain and final body weight.
The science is sound, experimental design and statistical analysis and conclusions made are appropriate.
The reviewer has one comment:
Since there were numerous rabbits from each breed originally included in the study, could the authors not have sampled at more time points during the experiment?
Response: our study aimed at comparing the microbial populations in jejunum and cecum of two rabbit breeds different in growth: Native Middle-Egypt Breed (NMER) and Giant Flander (GF). So, we focused on 12 weeks as a fixed time point to get results about the following questions: 1) Are the microbial populations in jejunum and cecum different? 2) Are microbial populations in two breeds of rabbits NMER and GF different? So we think that fixing the age is required and we chose 12 weeks of age since it is considered as a standard age for studying the growth period of the rabbit and the changes in gastrointestinal tracts are stable.
There are also minor English/Grammar corrections needed.
Response: We did our best to improve the English/Grammar in the text.
Reviewer 2 Report
Authors provided a comprehensive study about microbiota composition in two intestinal segments in two rabbit breeds. However, I have some concerns and questions and explain them in more detail in the attached file.

Minor modification in English can be made.
Author Response
Dear reviewer 2
Greetings for the day!
Greetings for the day!
Manuscript ID: animals- 2399147 “Title: Diversity and Co-occurrence Pattern Analysis of Cecal and Jejunal Microbiota in Two Rabbits Breeds”
Thank you and all the reviewers for pointing out those details.
Please below that we respond to your comments point to point.
Authors provided a comprehensive study about microbiota composition in two intestinal segments in two rabbit breeds. However, I have some concerns and questions and explain them in more detail in the attached file.
Peer review report on ”Diversity and Co-occurrence Pattern Analysis of Cecal and Jejunal Microbiota in Two Rabbits Breeds” by El-Sayed M. Abdel-Kafy, Kamel I. Kamel, Marco Severgnini, Shama H. A. Morsy, Paola Cremonesi, Shereen, S. Ghoneim, Gabriele Brecchia, Neama I. Ali, Yasmein Z. Abdel-Ghafar, Wael A.H.Ali, Hoda M.A. Shabaan.
General comments:
The authors examined the microbiota composition in two intestinal segments (cecum and jejunum) of two rabbit breeds (the Native Middle-Egypt Breed and Giand Flender) by 16S rRNA sequencing. Biochemical parameters, intestine histology, and electron microscopic measures were also examined in this study.
The topic of the paper is interesting, the chapters of the manuscript are well-structured, the study design is mostly fair, the paper is generally clear and well-written, and the results obtained can deliver useful information for the readers. However, I have some concerns and explain them in more detail below. I ask that the authors specifically address each of my comments in their responses.
Major comments:
Abstract:
- Authors mention „At 12 weeks of age, average body weight and most histological parameters were higher in GF than in NMER rabbits”. In my opinion, this statement is not true, since daily weight gain was higher and not body weight in the GF breed. On the other hand, only villus height was higher in the GF breed. Clarify in the text, please, and also mention what we can conclude from these results.
Response: We corrected these sentences and highlighted them in yellow color (lines 41-43).
- I am missing one-one sentences about methodology and results for biochemical parameters and histology/SEM results.
Response: We added two sentences in the Results section about biochemical parameters and histology/SEM results. We highlighted them in yellow color (lines 41-44).
Keywords:
- I would also add the exact breeds to the keywords.
Response: We added the NMER breed and the Giant Flander breed. We leave it to the Editor the final decision to accept them. Please notice the keywords list is limited from three to ten words.
Introduction:
- (lines 78-79): My question to the authors: Does it have any previous evidence that microbiome can alter in different breeds with different growth rates? Why is it important to examine these two breeds? Why can be the conclusion of this study beneficial? I would dedicate 1 or 2 sentences to these explanations at the end of the introduction chapter.
Response: We added some sentences to report what you requested (Lines 81-90) and highlighted them in yellow color.
- (lines 79-82) In my opinion, the last sentence of the introduction should be reworded and should be complemented with those examinations (histology, biochemical parameters, and SEM) that are also used in this study (why were those measured and what methods were used).
Response: We added and modified sentences to report what you requested (Line 88-90) and highlighted them by yellow color.
Materials and methods:
- Why were only male rabbits used in this study?
Response: The choice was made in order to avoid a possible effect of sex on the results; we reported that in lines 112-113 and highlighted by yellow color.
- There are many examinations in this study, which are described in subchapters. In my opinion, it would be very important to describe the number of individuals/samples used for the measures in every chapter.
Response: we added a number of animals and samples in every chapter and highlighted them in yellow color.
- In the histological subchapter, it would be useful to describe those intervals used to measure these parameters (villus length, width, etc.) and also mention a reference for this.
Response: we added a reference, highlighted by yellow color in line 136, for the histological study the readers can refer to for more details.
- How many villi, etc. were measured? It should be added to the text.
Response: we added a number of villi, etc. These adding were highlighted by yellow color.
- In the figures, there are no SD or SEM values. Add those and also a figure legend with more explanations (abbreviations, statistics, etc.)
Response: SEM values and statistics have been added to the figures, as requested.
Results:
- The subchapter „Body weight and body weight gain” is very short and p values are also missing. Add these to the text, please.
Response: We increased this subchapter and mentioned p-values in the text, as well as in Table 2.
- (lines 452-453) The authors mention, that the Lactobacillus order was more abundant in jejunum samples than in cecum and also mention the relative abundance. My concern is that this data (and others too) do not appear in the relevant table and the readers just suspect what are these numbers for. I suggest indicating these data somehow in tables, too.
Response: we added columns in Tables 3, 4, 5, and 6 to show the relative abundances at Order and Phylum levels, obtained by summing up the taxa at the lower levels (i.e.: genus) and highlighted by yellow color.
- Comparing the microbiota composition between organs resulted in more changes in the GF breed. What can be the main reason behind this?
Response: We discussed that in the Discussion part whereas; we discussed our results with other results, please read lines (608-621). Also, we added an explanation for the reason for the significant difference in the microbiota composition between organs in the GF breed (647-651). These parts are highlighted in yellow color.
Discussion:
- In the discussion chapter, I am missing the explanation of hematology and SEM results in the examined breeds and other rabbit breeds, too.
Response: please we added explanations for the significantly different in Total Protein and albumin that were higher in GF than in NM rabbits (Line 584-589). We added an explanation of SEM results in the examined breeds (Line 590-595). These parts are highlighted in yellow color.
Minor comments:
(line 23) „microbes” or „microorganisms” instead of „microbial agents”. (line 37) productive performance is word repetition.
Response: corrected as suggested and highlighted by gray color.
(lines 43-44): Beta-diversity of microbiota between jejunum and cecum was significant differences. This sentence should be rewritten.
Response: the sentence has been corrected as suggested and highlighted in gray color.
(line 106) NMER breed is sometimes abbreviated as NMER and NM. In my opinion, it’s very confusing, to choose one abbreviation at the beginning and use it throughout the entire manuscript, consistently.
Response: The Native Middle-Egypt Breed – NMER was abbreviated to NM at the beginning of Abstract and Material and Methods, whereas we abbreviated it as NM in all text.
(line 303) „Rabbit” is a word repetition.
Response: We removed it, following the reviewer’s suggestion
(line 450) „.. were significantly different between the examined organs”.
Response: corrected as suggested
(line 505) „comparisons between organs in GF breed” instead of samples.
Response: the sentence has been corrected as suggested
(line 708) in rabbits? Clarify in the text, please. Table 1. Some ingredients are written with capital letters and some are not. Write them consistently, please. Table 2. If abbreviations are used, explain them in the text and use abbreviations for all parameters (e.g. BW at week 5, DWG).
Response: The reviewer’s guess (line 708) is right: we are referring to rabbits. We changed the reference in order to correctly address this point and highlighted it in gray color.
Response: We have rewritten the letters a capital Table 1 and highlighted them in gray color.
Response: We did not use abbreviations in Table 2.
Reviewer 3 Report
The paper written by Abdel-Kafy et al evaluates the productive performance and microbiota variation in the jejunum and cecum of two rabbit breeds (NM and GF) with different growth rates. This study explains the relationship between growth speed and microbiota in cecum and jejunum, however, there are some issues in this study that need to be addressed.
Major comments:
1. The aminal number used in this study is confusing. For the blood biochemistry parameters, there are n=20 for each breed. Then in the description of the experimental design, it is 20 for NM and GF breeds. Finally, at the 16S Illumina sequencing part, it is 40 including both cecum and jejunum samples. If you are using 20 rabbits in total, then 40 is the right number. But if it is n=20 for each breed, then it is the wrong number. Also, if you used different rabbits for 16S rRNA sequencing and blood biochemical tests, then they are not comparable. Thus the authors need to address this issue.
2. The statistical analysis in this study is not proper. The authors should use one-way ANOVA to do the statistical analysis, not t-test, based on the data. Also, the authors need to put the error bars in Figures 1 and 2.
Minor comments:
1. Line 58-63: Too long sentence, need to be re-write.
2. Line 71: incomplete sentence.
3. Slaughter is not a proper word for animal research, euthanize is a better word for describing animal research.
4. Line 151-152: The website cannot be opened.
5. Figure 3: The sequence of the figures is not friendly to readers, also the authors need to present the scale bars on the figures to give clear guidance to readers.
6. Microbial profile: How about the data size in the cecum compare to the jejunum?
7. The characters in some figures are too small to read.
Author Response
Dear reviewer 3
Greetings for the day!
Manuscript ID: animals- 2399147 “Title: Diversity and Co-occurrence Pattern Analysis of Cecal and Jejunal Microbiota in Two Rabbits Breeds”
Thank you and all the reviewers for pointing out those details.
Please below that we respond to your comments point to point.
The paper written by Abdel-Kafy et al evaluates the productive performance and microbiota variation in the jejunum and cecum of two rabbit breeds (NM and GF) with different growth rates. This study explains the relationship between growth speed and microbiota in cecum and jejunum, however, there are some issues in this study that need to be addressed.
Major comments:
- The aminal number used in this study is confusing. For the blood biochemistry parameters, there are n=20 for each breed. Then in the description of the experimental design, it is 20 for NM and GF breeds. Finally, at the 16S Illumina sequencing part, it is 40 including both cecum and jejunum samples. If you are using 20 rabbits in total, then 40 is the right number. But if it is n=20 for each breed, then it is the wrong number. Also, if you used different rabbits for 16S rRNA sequencing and blood biochemical tests, then they are not comparable. Thus the authors need to address this issue.
Response: we corrected "n=20 for each breed to n=10 for each breed (line 120) and highlighted it in yellow color.
- The statistical analysis in this study is not proper. The authors should use one-way ANOVA to do the statistical analysis, not t-test, based on the data. Also, the authors need to put the error bars in Figures 1 and 2.
Response: we corrected this mistake in line 236 and highlighted it in yellow color. Also, we added error bars in Figures 1 and 2 and the values.
Minor comments:
- Line 58-63: Too long sentence, need to be re-write.
Response: The sentence has been re-written (lines 60-67)
- Line 71: incomplete sentence.
Response: we completed it (lines 74-75).
- Slaughter is not a proper word for animal research, euthanize is a better word for describing animal research.
Response: We used "euthanize" and highlighted by gray color
- Line 151-152: The website cannot be opened.
Response: The link was indeed broken since it contained erroneously a white space. We have modified the text and link pointing to the correct resource and highlighted in gray color
- Figure 3: The sequence of the figures is not friendly to readers, also the authors need to present the scale bars on the figures to give clear guidance to readers.
Response: we modified it and we added scale bars on the figures.
- Microbial profile: How about the data size in the cecum compare to the jejunum?
The number of z OTUs for the two intestinal tracts, for each breed, was comparable, as reported in the following table:
|
|
Jejunum, GF breed |
Jejunum, NM breed |
Cecum, GF breed |
Cecum, NM breed |
|
Mean |
4731.3 |
3901.7 |
5474.8 |
4824.3 |
|
Standard deviation |
2016.2 |
1969.0 |
1502.2 |
1636.6 |
Comparing the number of zOTUs between breeds and intestinal tracts by means of an unpaired t-test revealed no significant difference (p>0.05). However, if the reviewer was referring to something else, we courteously ask him/her to be more precise on the request, in order to give a proper answer to it.
- The characters in some figures are too small to read.
Response: We modified it and we added scale bars on the figures
Round 2
Reviewer 3 Report
Most of the issues have been addressed. However, the error bars in Figs. 1 and 2 cannot present how the individual observations are dispersed around the average. The authors need to calculate the standard deviation represents the amount of dispersion of the variable. Calculated as the root square of the variance. Not the SEM error bars, which are harder to interpret than a confidence interval.
Author Response
Dear Reviewer 2
Greetings for the day!
We are pleased to submit the revised-2 -manuscript ID: animals- 2399147 “Title: Diversity and Co-occurrence Pattern Analysis of Cecal and Jejunal Microbiota in Two Rabbits Breeds”
Thank you.
Please below we response to your comment.
Reviewer 2
Most of the issues have been addressed. However, the error bars in Figs. 1 and 2 cannot present how the individual observations are dispersed around the average. The authors need to calculate the standard deviation represents the amount of dispersion of the variable. Calculated as the root square of the variance. Not the SEM error bars, which are harder to interpret than a confidence interval.
Response: SD values have been added to Figs. 1 and 2, as requested.